# Representativeness of a mobile phone-based coverage evaluation survey following mass drug administration for soil-transmitted helminths: a comparison of participation between two cross-sectional surveys

Rohan Michael Ramesh ![ORCID],[1] William E Oswald ![ORCID],[2,3] Gideon John Israel,[1] Kumudha Aruldas,[1] Sean Galagan,[4] Hugo Legge,[2] Saravanakumar Puthupalayam Kaliappan,[1] Judd Walson ![ORCID],[4] Katherine E Halliday ![ORCID],[2] Sitara S R Ajjampur ![ORCID] [1]

RMR and WEO are joint first authors.

For numbered affiliations see end of article.

**Correspondence to**
Dr Sitara S R Ajjampur;
sitararao@cmcvellore.ac.in

## ABSTRACT

**Objectives** With increasing mobile phone subscriptions, phone-based surveys are gaining popularity with public health programmes. Despite advantages, systematic exclusion of participants may limit representativeness. Similar to control programmes for neglected tropical diseases (NTDs), the DeWorm3 trial of biannual community-wide mass drug administration (MDA) for elimination of soil-transmitted helminth infection used in-person coverage evaluation surveys to measure the proportion of the at-risk population treated during MDA. Due to lockdown during the COVID-19 pandemic, a phone-based coverage evaluation survey was necessary, providing an opportunity for the current study to compare representativeness and implementation (including non-response) of these two survey modes.

**Design** Comparison of two cross-sectional surveys.

**Setting** The DeWorm3 trial site in Tamil Nadu, India, includes Timiri, a rural subsite, and Jawadhu Hills, a hilly, hard-to-reach subsite inhabited predominantly by a tribal population.

**Participants** In the phone-based and in-person coverage evaluation surveys, all individuals residing in 2000 randomly selected households (50 in each of the 40 trial clusters) were eligible to participate. Here, we characterise household participation.

**Results** Of 2000 households, 1780 (89.0%) participated during the in-person survey. Of 2000 households selected for the phone survey, 346 (17.3%) could not be contacted as they had not provided a telephone number during the census and 1144 (57.2%) participated. Smaller households, households with lower socioeconomic status and those with older, women or less educated household-heads were under-represented in the phone-based survey compared with censused households. Regression analysis revealed non-response in the phone-based survey was higher among households from the poorest socioeconomic quintile (prevalence ratio (PR) 2.3, 95% CI 2.0 to 2.7) and lower when heads of households had completed secondary school or higher education (PR 0.7, 95% CI 0.6 to 0.8).

**Conclusions** Our findings suggest phone-based surveys under-represent households likely to be at higher risk of NTDs and in-person surveys are more appropriate for measuring MDA coverage within programmatic settings.

**Trial registration number** NCT03014167.

## STRENGTHS AND LIMITATIONS OF THIS STUDY

⇒ A detailed and up-to-date census in the study population allowed us to robustly assess representativeness of both survey modes.
⇒ The two surveys were conducted 6 months apart using the same questionnaire, allowing us to compare participation confidently.
⇒ Generalisability may be limited as the study was conducted in a setting with high mobile phone usage and network coverage, and during the COVID-19 pandemic and resulting lockdown.

## INTRODUCTION

Telephone-based surveys have been widely used in public health programmes and research, for purposes such as determining immunisation coverage of children under 3 years, conducting surveillance for risk factors related to leading causes of death, assessing mental health and self-reporting alcohol consumption.[1–5] This survey mode has become popular with increasing mobile phone subscription rates in low and middle-income countries, where cell phone coverage in some regions compares with that in high-income countries.[6 7] Cost-effectiveness, time efficiency, increased reach and willingness to

share sensitive information are some advantages of phone-based surveys.[8 9] Disadvantages of phone-based surveys include the inability to collect visual data, lower response rates in some communities and exclusion of participants without access to phones.[9 10] Mobile phone ownership is not universal, and the practical appeal of phone-based surveys may come at a cost by systematically excluding segments of the population, raising questions about the representativeness of the estimates they yield.[8 11–13] India's average tele-density, or the number of telephone connections (wireless and wired) for every hundred individuals living within an area, is 88.5, with a slight disparity between most rural and urban areas.[14 15] Tamil Nadu, in southern India, with a population of 72 million, is the fifth most tele-dense state of the 28 states in India with 108.5 telephone connections per 100 people, which is higher than the Indian average, with high mobile phone penetration even in rural households.[14 15]

Neglected tropical diseases (NTDs) predominantly affect impoverished, rural communities, and preventive chemotherapy (PC) through mass drug administration (MDA) is one of the key public health interventions implemented to control soil-transmitted helminth (STH) infections, lymphatic filariasis (LF), trachoma, schistosomiasis and onchocerciasis.[16] The WHO refers to coverage evaluation surveys as population-based surveys that offer a simple and effective method to accurately assess programme performance and provide precise estimates of PC coverage, or the proportion of individuals who swallowed the medicine or combination of medicines, for targeted NTDs.[17 18] These surveys are a valuable tool for evaluating the performance of NTD control programmes. For programmes targeting LF, trachoma and onchocerciasis, community-wide MDA (cMDA) coverage is assessed by household surveys using a standardised WHO tool.[17 19]

DeWorm3 is a trial of cMDA for interrupting transmission of STH, during which household coverage evaluation surveys were conducted after each of six cMDA rounds in the years 2018–2020.[20 21] In April 2020, however, during the nationwide lockdown imposed by the Government of India as a control measure during the COVID-19 pandemic, a phone-based survey was conducted instead of an in-person survey as in the previous four rounds. Leveraging this change in the trial protocol, necessitated by the COVID-19 pandemic, we conducted this analysis to contrast the representativeness, or household participation, and implementation of a phone-based survey with a previous in-person survey conducted 6 months earlier.

## METHODS
### DeWorm3 trial

DeWorm3 is a cluster randomised-controlled, community-based intervention trial that aims to determine the feasibility of interrupting STH transmission in Benin, India and Malawi.[21 22] In brief, 40 clusters at each study site were randomised to control clusters, to receive the standard of care, which in India is two times yearly deworming of children attending schools and preschools between 1 and 19 years of age during National Deworming Days (NDDs),[23] or intervention clusters, to receive two times yearly cMDA (delivered door-to-door) for 3 years (between 2018 and 2020) (figure 1).[24] At the start of each of the 3 years, an annual census was conducted to obtain accurate population data at the individual and household level and optimise coverage estimates during cMDAs. In all intervention clusters, cMDA was implemented simultaneously the day after NDD, followed by a mop-up revisiting previously unavailable households. Following each of the six rounds of cMDA, coverage evaluation surveys were conducted in the intervention and control clusters within a week of cMDA. Surveys in control clusters were intended to measure the coverage of treatment provided through the NDD programme. All census and coverage evaluation surveys (in-person and phone-based) were collected using the same electronic data collection forms programmed into the SurveyCTO mobile application (Dobility; Cambridge, Massachusetts and Washington, DC) run on encrypted Android smartphones. The survey forms in this application were embedded with logic checks, skip logic, numerical constraints and pre-loaded with location and demographic information to reduce errors while entering data.

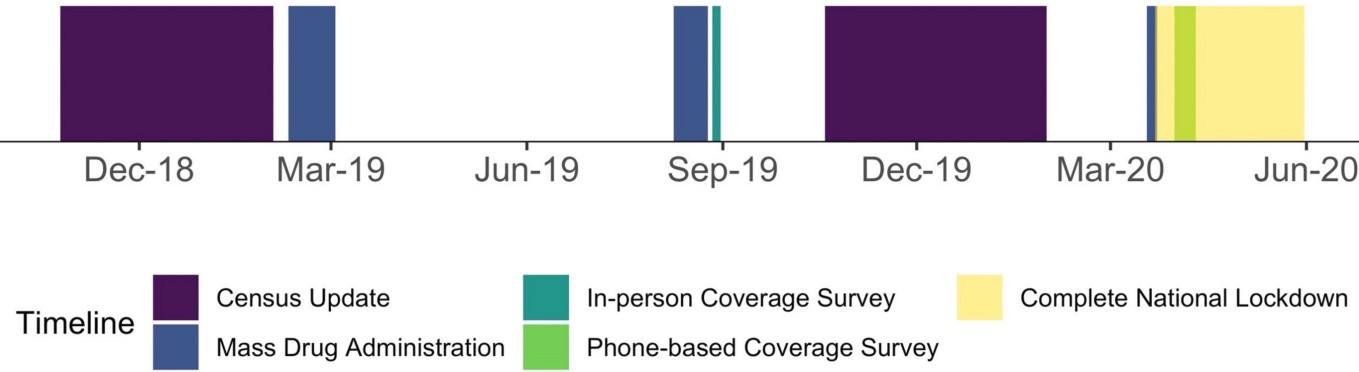

**Figure 1** Timeline of phone-based coverage evaluation survey in relation to other key activities.

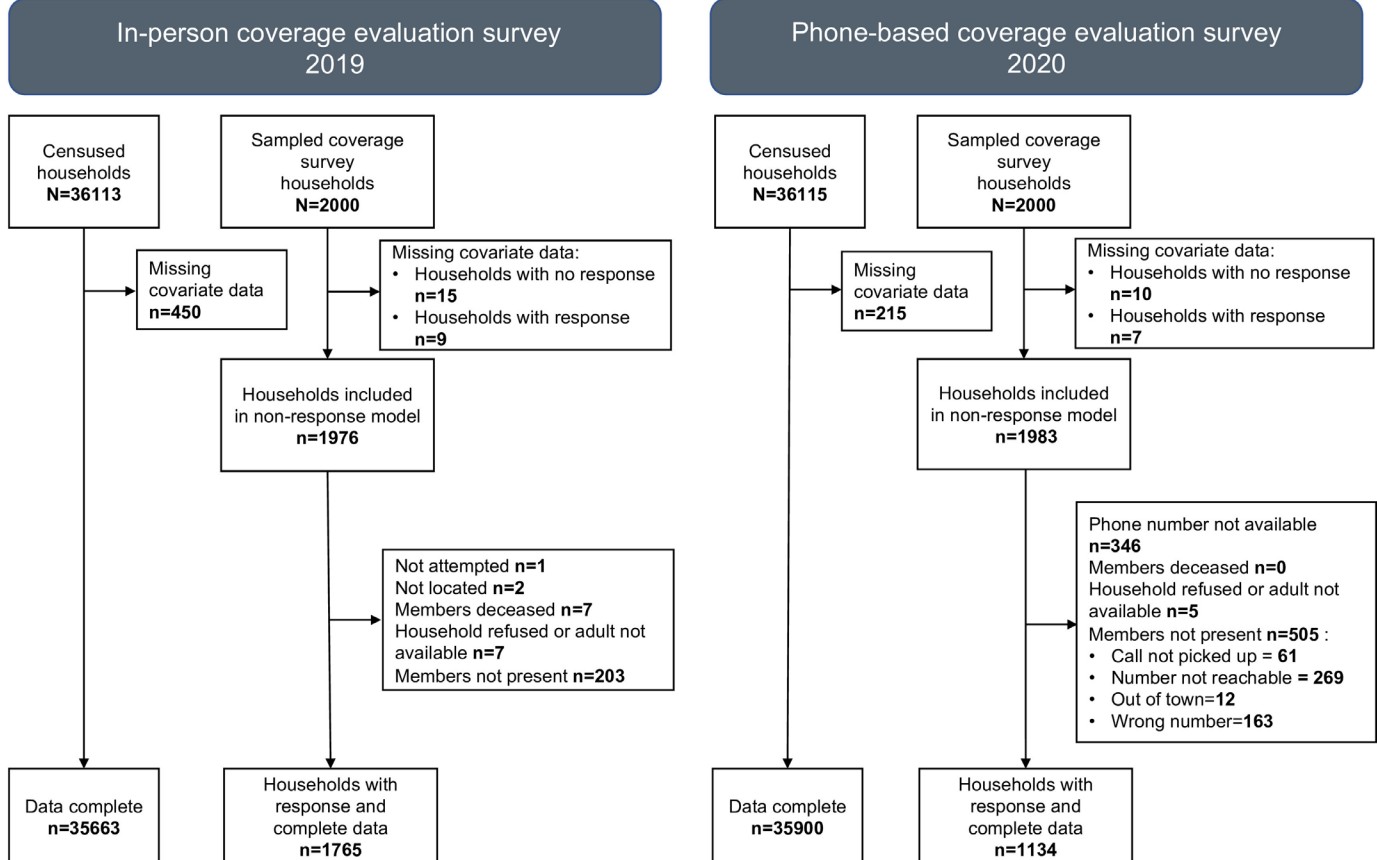

**Figure 2** Household participation details of in-person and phone-based coverage surveys.

## Study setting

The DeWorm3 trial site in India includes a subsite in Timiri, a rural area in Ranipet district (formerly a part of Vellore district), and another subsite in Jawadhu Hills in Tiruvannamalai district, a hilly, hard-to-reach area inhabited predominantly by tribal groups. Both subsites are located in the southern state of Tamil Nadu. The study covers an annually enumerated population of nearly 141 000, residing in approximately 37 000 households, with a total area of 477 km$^2$.[20] Men and women are equally represented, and children below 15 years of age constitute 23% of the population. Nearly a third of the adults received no formal education, with 20% and 90% of the population engaged in agricultural activities in Timiri and Jawadhu Hills, respectively.[21] More details on India's study site and population have been published previously.[20]

## Coverage evaluation surveys

For the coverage evaluation surveys, 50 households were randomly selected to participate from each of the 40 clusters (n=2000). Questions in the survey aligned with the WHO-endorsed coverage evaluation survey for use by national programmes, including individual-level questions on receipt and swallowing of the drug and reasons for not taking treatment, if applicable (online supplemental file).[18] The in-person coverage evaluation survey in 2019 was conducted from 27 to 31 August, based on a household sampling frame derived from the previous annual

census (25 October 2018 to 2 February 2019) (figure 1). All households had to be visited up to three times before they were considered unavailable, and the reasons were recorded. Sampled households that were not located or unavailable were substituted as necessary with households from two randomly selected staged replacement lists of 20 households per cluster. First, the respondent answered the household-level questions, and then each member present was interviewed. Proxy responses from household members were accepted for children under 5 years of age and household members absent on the second or third visit.

The phone-based coverage evaluation survey was conducted from 31 March to 10 April 2020 with a household sampling frame derived from the previous annual census (19 October 2019 to 31 January 2020) (figure 1). All households were eligible for selection, regardless of whether they had provided a phone number during the census or not. Households without a telephone number, who did not consent, or whose number was not reachable (two attempts made 4 hours apart were made on the first day and the third and fourth attempts made the next day) were also substituted with households from replacement lists. The interviewers were trained to administer the coverage evaluation survey over the phone while recording responses on the SurveyCTO mobile application on another device. The next available adult was

**Table 1** Comparison of household characteristics between censused and participating households for in-person and phone-based coverage surveys in Vellore, India, 2018–2020

| | Census households (2018–2019) | In-person coverage survey | | Census households (2019–2020) | Phone-based coverage survey | |
|---|---|---|---|---|---|---|
| | N=35 663 n (%) | N=1765 n (%) | P-value* | N=35 900 n (%) | N=1134 n (%) | P-value* |
| *Site details* | | | | | | |
| Study arm | | | | | | |
| Control | 17 420 (48.8) | 880 (49.9) | 0.41 | 17 431 (48.6) | 564 (49.7) | 0.43 |
| Intervention | 18 243 (51.2) | 885 (50.1) | | 18 469 (51.4) | 570 (50.3) | |
| Study subsite | | | | | | |
| Timiri | 27 638 (77.5) | 1394 (79.0) | 0.14 | 27 959 (77.9) | 892 (78.7) | 0.53 |
| Jawadhu Hills | 8025 (22.5) | 371 (21.0) | | 7941 (22.1) | 242 (21.3) | |
| *Household characteristics* | | | | | | |
| Religion | | | | | | |
| Other | 1086 (3.0) | 45 (2.5) | 0.24 | 1065 (3.0) | 41 (3.6) | 0.21 |
| Hindu | 34 577 (97.0) | 1720 (97.5) | | 34 835 (97.0) | 1093 (96.4) | |
| Caste | | | | | | |
| Higher caste | 404 (1.1) | 17 (1.0) | 0.66 | 410 (1.1) | 14 (1.2) | 0.55 |
| Backward caste | 10 667 (29.9) | 536 (30.4) | | 10 727 (29.9) | 364 (32.1) | |
| Most backward caste | 8533 (23.9) | 440 (24.9) | | 8720 (24.3) | 274 (24.2) | |
| Scheduled caste | 7960 (22.3) | 393 (22.3) | | 8018 (22.3) | 241 (21.3) | |
| Scheduled tribes | 8099 (22.7) | 379 (21.5) | | 8025 (22.4) | 241 (21.3) | |
| House type† | | | | | | |
| Concrete | 18 525 (51.9) | 896 (50.8) | 0.78 | 19 001 (52.9) | 637 (56.2) | 0.13 |
| Mixed | 3975 (11.1) | 203 (11.5) | | 3957 (11.0) | 120 (10.6) | |
| Government donated/funded house | 3380 (9.5) | 167 (9.5) | | 3318 (9.2) | 105 (9.3) | |
| Thatched | 9783 (27.4) | 499 (28.3) | | 9624 (26.8) | 272 (24.0) | |
| Socio-economic quintiles | | | | | | |
| Least poor quintile | 7120 (20.0) | 343 (19.4) | 0.58 | 7151 (19.9) | 312 (27.5) | <0.01 |
| Fourth quintile | 6965 (19.5) | 321 (18.2) | | 7179 (20.0) | 229 (20.2) | |
| Third quintile | 7273 (20.4) | 375 (21.2) | | 7134 (19.9) | 221 (19.5) | |
| Second quintile | 7118 (20.0) | 363 (20.6) | | 7212 (20.1) | 195 (17.2) | |
| Poorest quintile | 7187 (20.2) | 363 (20.6) | | 7224 (20.1) | 177 (15.6) | |
| Large household (family size) | | | | | | |
| No (≤4 members) | 24 027 (67.4) | 1180 (66.9) | 0.65 | 23 679 (66.0) | 683 (60.2) | <0.01 |
| Yes (≥5 members) | 11 636 (32.6) | 585 (33.1) | | 12 221 (34.0) | 451 (39.8) | |
| *Head of the household characteristics* | | | | | | |
| Age | | | | | | |
| 18–30 years | 2143 (6.0) | 92 (5.2) | 0.68 | 1516 (4.2) | 53 (4.7) | <0.01 |
| 31–40 years | 7071 (19.8) | 348 (19.7) | | 6842 (19.1) | 246 (21.7) | |
| 41–50 years | 9103 (25.5) | 451 (25.6) | | 9192 (25.6) | 331 (29.2) | |
| 51–60 years | 7827 (21.9) | 402 (22.8) | | 8417 (23.4) | 247 (21.8) | |
| >61 years | 9519 (26.7) | 472 (26.7) | | 9933 (27.7) | 257 (22.7) | |
| Sex | | | | | | |
| Male | 28 260 (79.2) | 1412 (80.0) | 0.44 | 29 198 (81.3) | 964 (85.0) | <0.01 |
| Female | 7403 (20.8) | 353 (20.0) | | 6702 (18.7) | 170 (15.0) | |
| Education level | | | | | | |
| No education | 12 047 (33.8) | 588 (33.3) | 0.55 | 10 994 (30.6) | 270 (23.8) | <0.01 |
| Any primary | 7604 (21.3) | 379 (21.5) | | 8161 (22.7) | 256 (22.6) | |

Continued

**Table 1** Continued

| | Census households (2018–2019) | In-person coverage survey | | Census households (2019–2020) | Phone-based coverage survey | |
|---|---|---|---|---|---|---|
| | N=35663 n (%) | N=1765 n (%) | P-value* | N=35900 n (%) | N=1134 n (%) | P-value* |
| Any middle | 6210 (17.4) | 329 (18.6) | | 6419 (17.9) | 202 (17.8) | |
| Any secondary or higher | 9802 (27.5) | 469 (26.6) | | 10326 (28.8) | 406 (35.8) | |
| Marriage status | | | | | | |
| Never married | 6778 (19.0) | 324 (18.4) | 0.50 | 7446 (20.7) | 168 (14.8) | <0.01 |
| Ever married | 28885 (81.0) | 1441 (81.6) | | 28454 (79.3) | 966 (85.2) | |
| Owns mobile phone | | | | | | |
| No | 5243 (14.7) | 268 (15.2) | 0.75 | 5430 (15.1) | 43 (3.8) | <0.01 |
| Yes | 27972 (78.4) | 1382 (78.3) | | 30470 (84.9) | 1091 (96.2) | |
| Data not available | 2448 (6.9) | 115 (6.5) | | | | |

*Pearson's $\chi^2$ test.
†Concrete—concrete walls and roof; Mixed—concrete walls and tiled roof; Government donated/funded house—prebuilt government houses or houses funded through schemes for economically and socially marginalised groups; Thatched—thatched walls and roof.

interviewed using the same phone number if the head of the household (preferred respondent) was unavailable for three attempts. Thereafter, other household members were also interviewed over the phone during the same call. The primary respondent was allowed to provide proxy responses for children under 5 years and household members absent on any attempt.

## Data analysis

To assess representativeness, we compared the characteristics of households that participated in each of the coverage evaluation surveys with those of households in the previous census not randomly selected for the coverage evaluation survey and identified factors associated with household non-response during both the coverage evaluation surveys. To achieve the sample size for each coverage evaluation survey, additional lists of households were randomly selected and released to the survey teams to compensate for unavailable households. For the current study, contrasting representativeness, we included only the initial 2000 households sampled for each survey and restricted analyses to households with complete data. We compared characteristics of households participating in each of the coverage evaluation surveys with households in the most recent previous annual census considering age, sex, reported education and marital status of household head; household religion and caste and period of time in residence and house type. Principal component analysis using household assets was used to arrive at a composite wealth index and divided into five socioeconomic status (SES) quintiles, as described previously.[20] Participants who were currently married, separated, widowed or divorced were considered ever married. $\chi^2$ tests were used to compare distributions of household characteristics between census and surveyed households.

For each coverage evaluation survey, we identified factors associated with household non-response by fitting modified Poisson regression models with robust errors to adjust for clustering.[25] For this analysis, a household was considered to have a response if at least one household members were present and agreed to respond to the survey. Households not meeting this criterion for any reason were considered a non-response. We estimated the prevalence ratio (PR) between each candidate predictor variable and non-response. We then used a best-subset selection approach, modelling all possible combinations of candidate predictor variables hypothesised to influence a household's participation and selecting the final model with the smallest Bayesian information criterion.[26] All candidate models included subsite, and household caste was excluded from the multivariable model selection because of collinearity with subsite. In a sensitivity analysis, we examined the inclusion of mobile phone ownership and phone number availability as additional candidate predictors to determine factors associated with household non-response. As a *post hoc* analysis, we also examined the association between households who reported owning a mobile phone and phone number availability.

In addition, we contrasted the implementation of each of the coverage evaluation surveys by comparing the number of attempts to contact (visits or calls), time of the day the visits/calls were made, duration per visit or call, number of individuals interviewed per attempt, proportion of household residents interviewed per attempt, total duration to complete the household's survey and number of visits or calls to complete the household survey. We used a non-parametric K-sample test of medians for continuous measures, a Pearson's $\chi^2$ test for categorical measures, and compared mean total duration weighted by household size using a Wald test. Statistical significance for all tests was set at p value <0.05. The data were managed and analysed using STATA V.16.1 software (StataCorp, Texas).

**Table 2** Factors associated with non-response in a phone-based coverage evaluation survey in Vellore, India, 2020

| | Households* | Non-response† | Univariate | | Multivariable | |
|---|---|---|---|---|---|---|
| | N=1983 (100) n (%) | n=849 (42.8) n (%) | PR (95% CI) | P-value | PR (95% CI) | P-value |
| *Site details* | | | | | | |
| Proportion of households in each arm | | | | | | |
| Control | 993 (50.1) | 429 (43.2) | REF | 0.78 | | |
| Intervention | 990 (49.9) | 420 (42.4) | 1.0 (0.9 to 1.1) | | | |
| Proportion of households per site | | | | | | |
| Timiri | 1586 (80.0) | 694 (43.8) | REF | 0.45 | REF | <0.01 |
| Jawadhu Hills | 397 (20.0) | 155 (39.0) | 0.9 (0.7 to 1.2) | | 0.6 (0.4 to 0.8) | |
| *Household characteristics* | | | | | | |
| Religion | | | | | | |
| Other | 68 (3.4) | 27 (39.7) | REF | 0.67 | | |
| Hindu | 1915 (96.6) | 822 (42.9) | 1.1 (0.8 to 1.5) | | | |
| Caste | | | | | | |
| Higher caste | 20 (1.0) | 6 (30.0) | REF | 0.50 | – | – |
| Backward caste | 635 (32.0) | 271 (42.7) | 1.4 (0.8 to 2.6) | | – | – |
| Most backward caste | 487 (24.6) | 213 (43.7) | 1.5 (0.8 to 2.7) | | – | – |
| Scheduled caste | 439 (22.1) | 198 (45.1) | 1.5 (0.9 to 2.6) | | – | – |
| Scheduled tribes | 402 (20.3) | 161 (40.0) | 1.3 (0.7 to 2.5) | | – | – |
| House type‡ | | | | | | |
| Concrete | 1054 (53.2) | 417 (39.6) | REF | 0.01 | | |
| Mixed | 233 (11.7) | 113 (48.5) | 1.2 (1.1 to 1.4) | | | |
| Government funded/donated house | 200 (10.1) | 95 (47.5) | 1.2 (1.0 to 1.5) | | | |
| Thatched | 496 (25.0) | 224 (45.2) | 1.1 (1.0 to 1.3) | | | |
| Socio-economic quintiles | | | | | | |
| Least poor quintile | 422 (21.3) | 110 (26.1) | REF | <0.01 | REF | <0.01 |
| Fourth quintile | 390 (19.7) | 161 (41.3) | 1.6 (1.3 to 1.9) | | 1.5 (1.2 to 1.8) | |
| Third quintile | 406 (20.5) | 185 (45.6) | 1.7 (1.4 to 2.1) | | 1.6 (1.4 to 2.0) | |
| Second quintile | 386 (19.5) | 191 (49.5) | 1.9 (1.5 to 2.3) | | 1.9 (1.6 to 2.3) | |
| Poorest quintile | 379 (19.1) | 202 (53.3) | 2.0 (1.7 to 2.5) | | 2.3 (2.0 to 2.7) | |
| Large household (family size) | | | | | | |
| No (≤4 members) | 1286 (64.9) | 603 (46.9) | REF | <0.01 | | |
| Yes (≥5 members) | 697 (35.1) | 246 (35.3) | 0.8 (0.7 to 0.8) | | | |
| *Head of the household characteristics* | | | | | | |
| Age | | | | | | |
| 18–30 years | 84 (4.2) | 31 (36.9) | REF | <0.01 | | |
| 31–40 years | 388 (19.6) | 142 (36.6) | 1.0 (0.8 to 1.3) | | | |
| 41–50 years | 516 (26.0) | 185 (35.9) | 1.0 (0.8 to 1.2) | | | |
| 51–60 years | 431 (21.7) | 184 (42.7) | 1.2 (0.9 to 1.5) | | | |
| >61 years | 564 (28.4) | 307 (54.4) | 1.5 (1.1 to 2.0) | | | |
| Sex | | | | | | |
| Male | 1602 (80.8) | 638 (39.8) | REF | <0.01 | | |
| Female | 381 (19.2) | 211 (55.4) | 1.4 (1.2 to 1.6) | | | |
| Education level | | | | | | |
| No education | 587 (29.6) | 317 (54.0) | REF | <0.01 | REF | <0.01 |

Continued

**Table 2** Continued

| | Households* | Non-response† | Univariate | | Multivariable | |
|---|---|---|---|---|---|---|
| | N=1983 (100) n (%) | n=849 (42.8) n (%) | PR (95% CI) | P-value | PR (95% CI) | P-value |
| Any primary | 452 (22.8) | 196 (43.4) | 0.8 (0.7 to 0.9) | | 0.8 (0.7to 0.9) | |
| Any middle | 343 (17.3) | 141 (41.1) | 0.8 (0.6 to 0.9) | | 0.8 (0.6 to 0.9) | |
| Any secondary or higher | 601 (30.3) | 195 (32.4) | 0.6 (0.5 to 0.7) | | 0.7 (0.6 to 0.8) | |
| Marriage status | | | | | | |
| Never married | 397 (20.0) | 229 (57.7) | REF | <0.01 | | |
| Ever married | 1586 (80.0) | 620 (39.1) | 0.7 (0.6 to 0.8) | | | |

*Number (N) of sampled households with complete covariate data used to estimate prevalence of non-response in each category.
†Non-response—no household members available for interview, household refused to participate or unable to consent, or the household was not attempted.
‡Concrete—concrete walls and roof; Mixed—concrete walls and tiled roof; Government donated/funded house—prebuilt government houses or houses funded through schemes for economically and socially marginalised groups; Thatched—thatched walls and roof.
PR, prevalence ratio.

## Patient and public involvement
None.

## RESULTS

### Households available for participation in coverage evaluation surveys

During the 2019 in-person coverage evaluation survey, of the 2000 households sampled, 1780 (89.0%) participated, and during the 2020 phone-based coverage evaluation survey, 1144 (57.2%) of 2000 sampled households participated. Data were incomplete for 24 and 17 households of those sampled for the in-person and phone-based surveys, respectively. The number of households not participating due to various reasons during the in-person and phone-based surveys is illustrated in figure 2. During the in-person survey, members were unavailable in 203 (10.1%) households even after three visits. During the phone-based survey, 346 (17.3%) households could not be contacted as phone numbers were unavailable. Furthermore, 505 (25.2%) households were unavailable after three or more phone calls—these included 61 households, where the phone call was not picked up, 269 for whom the 'number not reachable' message was received from the phone service provider as the phone was outside an area of connection, 12 primary respondents informed the interviewers that they were 'out of town' and not present during the cMDA, and as a result, they were unable to answer on behalf of their family members, and 163 were recorded as 'wrong number' as the call was attended by persons not associated with the surveyed household.

### Representativeness of coverage evaluation surveys

No differences were observed when the characteristics of households and heads of households participating in the in-person coverage evaluation survey were compared with censused households that were not sampled (table 1). In the phone-based survey, however, participating households in the highest SES quintile were over-represented (27.5%) compared with non-sampled households (19.9%), with the inverse observed in the poorest quintile (p<0.01). Larger households with five or more household members were also over-represented in the phone-based survey (39.8%) compared with non-sampled households (34.0%, p<0.01).

When their characteristics were compared, heads of households participating in the phone-based survey were most commonly aged between 41 and 50 years (29.2%) in contrast to those in the non-sampled households who were mainly older than 60 years (27.7%, p<0.01). Similarly, households with female heads were under-represented in the phone-based survey (15.0%) compared with non-sampled households (18.7%, p<0.01). Households where the heads reported no formal education were also under-represented in the phone-based survey (23.8%) compared with non-sampled households (30.6%, p<0.01). Households headed by individuals who were never married were also significantly under-represented in the phone-based survey (14.8%) compared with non-sampled households (20.7%, p<0.01). Reported ownership of mobile phones was higher (96.2%) compared with non-sampled households (84.9%, p<0.01) in the phone-based coverage evaluation survey (table 1).

### Factors associated with household participation

Of the 1983 households with complete data sampled for the phone-based survey, 849 (42.8%) were considered non-responders. The selected modified Poisson regression model included SES and the education of the head of the household. Non-response increased with greater poverty, and households in the poorest quintile were more than two times as likely to have not responded as households in the least poor quintile (PR: 2.3, 95% CI 2.0 to 2.7), adjusting for site and education of the household head. Non-response was lower in households with heads who reported any education compared with those

**Table 3** Factors associated with non-response in an in-person coverage evaluation survey in Vellore, India, 2019

| In-person coverage survey | | | | | | |
|---|---|---|---|---|---|---|
| | Households* | Non-response† | Univariate | | Multivariable | |
| | N=1976 (100) n (%) | n=211 (10.7) n (%) | PR (95% CI) | P-value | PR (95% CI) | P-value |
| *Site details* | | | | | | |
| Proportion of households in each arm | | | | | | |
| Control | 990 (50.1) | 110 (11.1) | REF | 0.65 | | |
| Intervention | 986 (49.9) | 101 (10.2) | 0.9 (0.7 to 1.3) | | | |
| Proportion of households per site | | | | | | |
| Timiri | 1581 (80.0) | 187 (11.8) | REF | <0.01 | REF | <0.01 |
| Jawadhu Hills | 395 (20.0) | 24 (6.1) | 0.5 (0.3 to 0.8) | | 0.5 (0.3 to 0.8) | |
| *Household characteristics* | | | | | | |
| Religion | | | | | | |
| Other | 58 (2.9) | 13 (22.4) | REF | <0.01 | | |
| Hindu | 1918 (97.1) | 198 (10.3) | 0.5 (0.3 to 0.7) | | | |
| Caste | | | | | | |
| Higher caste | 18 (0.9) | 1 (5.6) | REF | 0.18 | – | – |
| Backward caste | 609 (30.8) | 73 (12.0) | 2.2 (0.3 to 16.0) | | – | – |
| Most backward caste | 502 (25.4) | 62 (12.4) | 2.2 (0.3 to 16.2) | | – | – |
| Scheduled caste | 439 (22.2) | 46 (10.5) | 1.9 (0.2 to 14.4) | | – | – |
| Scheduled tribes | 408 (20.6) | 29 (7.1) | 1.3 (0.2 to 9.8) | | – | – |
| House type‡ | | | | | | |
| Concrete | 1019 (51.6) | 123 (12.1) | REF | 0.02 | | |
| Mixed | 229 (11.6) | 26 (11.4) | 0.9 (0.6 to 1.4) | | | |
| Government funded/donated house | 191 (9.7) | 24 (12.6) | 1.0 (0.7 to 1.5) | | | |
| Thatched | 537 (27.2) | 38 (7.1) | 0.6 (0.4 to 0.9) | | | |
| Socio-economic quintiles | | | | | | |
| Least poor quintile | 391 (19.8) | 48 (12.3) | REF | 0.02 | | |
| Fourth quintile | 360 (18.2) | 39 (10.8) | 0.9 (0.6 to 1.3) | | | |
| Third quintile | 429 (21.7) | 54 (12.6) | 1.0 (0.7 to 1.6) | | | |
| Second quintile | 410 (20.7) | 47 (11.5) | 0.9 (0.6 to 1.5) | | | |
| Poorest quintile | 386 (19.5) | 23 (6.0) | 0.5 (0.3 to 0.8) | | | |
| Large household (family size) | | | | | | |
| No (≤4 members) | 1339 (67.8) | 159 (11.9) | REF | 0.02 | | |
| Yes (≥5 members) | 637 (32.2) | 52 (8.2) | 0.7 (0.5 to 0.9) | | | |
| *Head of the household characteristics* | | | | | | |
| Age | | | | | | |
| 18–30 years | 108 (5.5) | 16 (14.8) | REF | 0.03 | REF | <0.01 |
| 31–40 years | 403 (20.4) | 55 (13.6) | 0.9 (0.6 to 1.5) | | 0.8 (0.5 to 1.3) | |
| 41–50 years | 489 (24.7) | 38 (7.8) | 0.5 (0.3 to 0.9) | | 0.4 (0.3 to 0.7) | |
| 51–60 years | 455 (23.0) | 53 (11.6) | 0.8 (0.5 to 1.3) | | 0.6 (0.4 to 1.0) | |
| >61 years | 521 (26.4) | 49 (9.4) | 0.6 (0.3 to 1.2) | | 0.5 (0.3 to 0.9) | |
| Sex | | | | | | |
| Male | 1567 (79.3) | 155 (9.9) | REF | 0.02 | | |
| Female | 409 (20.7) | 56 (13.7) | 1.4 (1.1 to 1.8) | | | |
| Education level | | | | | | |
| No education | 655 (33.1) | 67 (10.2) | REF | 0.36 | | |

Continued

**Table 3** Continued

**In-person coverage survey**

| | Households* | Non-response† | Univariate | | Multivariable | |
|---|---|---|---|---|---|---|
| | N=1976 (100) n (%) | n=211 (10.7) n (%) | PR (95% CI) | P-value | PR (95% CI) | P-value |
| Any primary | 427 (21.6) | 48 (11.2) | 1.1 (0.8 to 1.5) | | | |
| Any middle | 360 (18.2) | 31 (8.6) | 0.8 (0.5 to 1.4) | | | |
| Any secondary or higher | 534 (27.0) | 65 (12.2) | 1.2 (0.9 to 1.6) | | | |
| Marriage status | | | | | | |
| Never married | 377 (19.1) | 53 (14.1) | REF | 0.01 | | |
| Ever married | 1599 (80.9) | 158 (9.9) | 0.7 (0.5 to 0.9) | | | |

*Number (N) of sampled households with complete covariate data used to estimate prevalence of non-response in each category.
†Non-response—no household members available for interview, household refused to participate or unable to consent, or the household was not attempted.
‡Concrete—concrete walls and roof; Mixed—concrete walls and tiled roof; Government donated/funded house—prebuilt government houses or houses funded through schemes for economically and socially marginalised groups; Thatched—thatched walls and roof.
PR, prevalence ratio.

who reported no education (p<0.01) in the phone-based survey (table 2), adjusting for site and SES. Of the 1976 households sampled for the in-person survey and with complete data, 211 (10.7%) were non-responders. Similar to the phone-based survey, the unadjusted estimate for non-response in the in-person survey was lower in the Jawadhu Hills subsite compared with the Timiri subsite (PR: 0.5, 95% CI 0.3 to 0.8), and age was selected (p<0.01) once subsite was included (table 3).

As a sensitivity analysis, we examined mobile phone ownership and the availability of phone numbers as additional candidate predictors. The final selected model for non-response in the phone-based survey included mobile phone ownership and SES, but it no longer included head of household education. Including mobile phone ownership attenuated but did not entirely remove the association of SES with non-response (households from the poorest SES quintile: PR: 1.7, 95% CI 1.4 to 2.1). Mobile phone ownership was strongly associated with non-response in the phone-based survey independent of SES and adjusting for subsite, but not in the in-person survey, for which the results did not change with the inclusion of mobile phone ownership as a candidate predictor (online supplemental tables 1,2). Using phone number availability in place of mobile phone ownership, the predictor was again selected and had a slightly stronger association with non-response in the phone-based survey model, but there was no change in results for the in-person survey (online supplemental tables 3,4). Cross-tabulation of phone number availability among those who reported owning a mobile phone showed that 93.5% (1461/1563) of the phone numbers were available in the in-person survey and decreased slightly to 89.6% (1524/1700) in the phone-based survey (online supplemental table 5).

In 2020, 5728 (15.1%) out of the 37883 censused households reported not owning a mobile phone. Households in the poorest quintile were 19 times more likely not to own a mobile phone than households in the least poor quintile (PR: 19.3, 95% CI 15.3 to 24.3). Non-ownership of phones was less likely in households with heads who were more educated compared with those who reported no education (p<0.01) (online supplemental table 6).

### Differences in the implementation of in-person and phone-based surveys

For the in-person survey, 2932 attempts were made to reach the 2000 households, during 2376 (81.0%) of which household respondents were located and available to participate. In the phone-based survey, 3262 attempts were made to reach the 2000 households, during which primary respondents were located and available to participate in 1166 (35.7%) attempts (table 4). The phone-based surveys were most often conducted after 18:00 hours, in contrast to the in-person interviews conducted mainly between 10:01 to 18:00 hours (p<0.01). The mean duration of each in-person survey visit was longer (3.5 min) than the mean time taken to complete a call during the phone-based survey (1.9 min, p<0.01). During the in-person survey, each visit interviewed a median of 2.0 individuals per household or 66.7 of residents compared with each call during the phone-based survey, when a median of 0 was interviewed per call (which includes all the calls where nobody was present/could be reached) (p<0.01).

Most in-person survey households were completed in one or two visits (97.0%), but by phone, households were mainly completed in the first attempt (68.8%). Considering the household size, the weighted total time to complete the household's survey was slightly longer during the in-person survey (9.3 min) than in the phone-based survey (8.8 min, p=0.05) (table 4).

**Table 4** Comparison of implementation of in-person and phone-based coverage surveys in Vellore, India, 2019–2020

|  | In-person coverage survey | Phone-based coverage survey | P-value* |
|---|---|---|---|
| *Discrete household visits or calls* | n=2932 (100.0) | n=3262 (100.0) | |
| Household status during visit or call | | | |
| Not located | 2 (0.1) | 1 (0.0) | |
| Members deceased | 7 (0.2) | 0 (0.0) | |
| Members not present | 547 (18.7) | 2095 (64.2) | <0.01 |
| Members present | 2376 (81.0) | 1166 (35.7) | |
| Time of day | | | |
| <10:00 | 149 (5.1) | 83 (2.5) | <0.01 |
| 10:01–12:00 | 656 (22.4) | 398 (12.2) | |
| 12:01–14:00 | 832 (28.4) | 785 (24.1) | |
| 14:01–16:00 | 659 (22.5) | 535 (16.4) | |
| 16:01–18:00 | 542 (18.5) | 457 (14.0) | |
| >18:00 | 94 (3.2) | 1004 (30.8) | |
| Mean duration per visit or call (min) | 3.5 (1.8–6.9) | 1.9 (1.0–4.4) | <0.01 |
| Median number of resident interviews completed per visit or call | 2.0 (1.0–4.0) | 0.0 (0.0–3.0) | <0.01 |
| Median proportion of resident interviews completed per visit or call (IQR) | 66.7 (25.0–100.0)† | 0.0 (0.0–100.0)† | <0.01 |
| *Household survey completion* | n=1765 (100.0) | n=1134 (100.0) | |
| Total duration to complete household survey (min) | 6.3 (3.6–10.6) | 6.5 (3.8–10.8) | 0.47 |
| Weighted mean total duration (min) ‡ | 9.3 (9.0–9.7) | 8.8 (8.4–9.2) | 0.05 |
| Visits or calls to complete household survey | | | |
| 1 | 981 (55.6) | 780 (68.8) | <0.01 |
| 2 | 730 (41.4) | 174 (15.3) | |
| 3 | 51 (2.9) | 61 (5.4) | |
| 4+ | 3 (0.2) | 119 (10.5) | |

*Pearson's $\chi^2$/T-test as appropriate. For status analysis, not located and members deceased were excluded due to small cell counts.
†Median (IQR).
‡Weighted according to the household size.

## DISCUSSION

In this study, we compared representativeness and implementation of a phone-based survey with an in-person survey to measure treatment coverage following cMDA for STH as part of the DeWorm3 trial in rural communities in southern India. Household participation in surveys assessing the coverage of the STH programme was higher when conducted in person than when done using a phone-based survey (89.0% vs 57.2%). When comparing the characteristics of participating households, larger households, households with higher SES and those with younger, more educated and male heads of household were over-represented in the phone-based survey. Adjusting for subsite, non-response in the phone-based survey was higher among households from the poorest SES quintile and households with uneducated household heads, while non-response in the in-person survey was associated with the age of the household head.

We observed that households with higher SES were over-represented in the phone-based survey. Studies conducted in other low and middle-income countries have demonstrated that households with higher SES are more likely to own mobile phones.[27 28] Furthermore, the phone-based survey was implemented during the COVID-19 lockdown in India when income and employment were drastically reduced, potentially increasing the challenges faced by poorer households to recharge phone subscriptions.[29 30] Both issues could likely explain lower overall participation rates among the poorest households in the phone-based survey, who were approximately half as likely to respond as the least poor households, based on the non-response model. Larger households were over-represented in the phone-based survey, potentially because the probability of someone in the family owning a phone may be higher in such households, but this characteristic was not selected during modelling. Similar to our findings related to SES, studies in Germany and the UK also reported an under-representation of socially disadvantaged and/or low-income participants in phone-based surveys.[8 10]

We also noted that participation of households headed by individuals in certain demographic groups differed between in-person and phone-based surveys. The phone-based survey had a higher proportion of respondents from households with younger, more educated and male heads of household compared with the censused population. The effect of education was also noted in our model revealing that non-response in the phone-based survey was lower in households with heads who reported any education compared with those who reported no education. The effect of age on participation in phone-based surveys has been well documented in related health surveys from the USA, Ireland, UK and Brazil.[8 12 13] In terms of gender, low phone ownership among women residing in South Asia has been reported previously due to traditional gender norms, technical illiteracy in operating a phone, male dominance over phone usage and economic reasons.[31]

Of the 1983 sampled households with complete data in the phone-based survey, 1685 (85.0%) had provided a mobile phone number during the annual census. The availability of a phone number does not always result in a response, however, often due to poor network coverage, unrenewed subscriptions or changed subscriptions, as reported in a study on cell phone ownership in Burkina Faso.[32] From the starting sample of 2000 households, nearly two times as many households participated in the in-person survey compared with the phone-based survey. If information on population characteristics is available beforehand, researchers conducting phone-based surveys could consider quota sampling, where participants are selected non-randomly according to a fixed quota or percentage of the population based on one or more characteristics, or post-stratification weight adjustment following data collection to ensure that all age groups and socioeconomic strata are equally represented.[10 33] Both methods can be employed if non-response remains high following quota sampling.

Interviewers initially contacted participants on the phone during the same working hours as the in-person survey. However, during implementation, we observed two distinct periods (mid-day between 12:01 to 14:00 hours and after 18:00 hours) of response in the phone-based survey (data not shown) and adapted the time of the calls to this observed pattern. Phone calls can occur in unexpected instances, and the owner may not be in a suitable environment to answer, such as during working hours, which can impact response rates.[34] Although phone-based surveys allowed us to adapt interview timings, we recommend considering the local culture and work patterns of the target population during planning to ensure high response rates. As observed in our study, phone-based surveys provide the advantage of flexibility in terms of survey timing, enabling data collection beyond the typical working hours and daylight constraints associated with in-person surveys, particularly in the context of safety considerations in many countries.

The in-person survey took marginally longer to complete as it necessitated the interviewer to verify and interview all available household members individually. This observed difference in duration would increase if the travel time to the house for an in-person survey, particularly in hard-to-reach rural areas, is considered. Arguably, the phone-based survey was faster to complete since one participant could respond on behalf of those absent, explaining higher completion rates on the first attempt. However, the number of attempts to establish a successful contact was higher in the phone-based survey because phone calls often failed to connect, similar to phone-based surveys conducted in Ireland and the UK.[8]

NTDs are more prevalent in socially and economically disadvantaged groups living in poor environmental conditions.[35] In this population, STH infections, particularly hookworm, are more prevalent in households belonging to poorer SES quintiles with less-educated household heads.[20] MDA programmes are one of the key public health interventions implemented to control or eliminate NTDs, including STH, and a coverage survey following an MDA programme evaluates the performance of the intervention. Therefore, it is imperative that the population included in a coverage evaluation survey does not exclude the same groups at a higher risk of infection to avoid biasing estimates of treatment coverage. In this regard, the phone-based survey under-represents many of these groups. Our findings suggest that in-person surveys better represent the target population in rural communities than phone-based surveys.

There are several advantages and limitations to this analysis. Our study within a trial setting provided the opportunity to compare both survey modes with exhaustive and up-to-date census information, allowing us to robustly assess the representativeness of both survey strategies. This analysis also offered an advantage over others based in low and middle-income countries since it was conducted in a rural setting with regular community engagement and used the same questionnaire in both surveys within a 6-month time frame, thus allowing us to compare participation confidently.[36–38] The phone-based survey was conducted in a state with high teledensity and fair network coverage, however, even in rural areas. As a result, these findings may not be generalisable to all rural settings. In addition, this survey was conducted during the COVID-19 pandemic and during a time of lockdowns in the study region, which may have negatively impacted responsiveness to the phone survey. Our exploration of representativeness and characteristics associated with non-response included a range of measures collected during the trial census, but there may be other unmeasured factors that we have not accounted for in our analyses.

## CONCLUSIONS

We found that phone-based surveys under-represent groups at a higher risk of acquiring STH and other NTDs.

As a result, phone surveys may bias coverage estimates and could potentially negatively impact the groups that most need treatment. Phone-based surveys are a convenient alternative, especially when circumstances preclude household visits and resources are limited, but in the absence of accurate, up-to-date, data on population characteristics, in-person surveys appear more representative.

**Author affiliations**
[1]The Wellcome Trust Research Laboratory, Division of Gastrointestinal Sciences, Christian Medical College Vellore, Vellore, Tamil Nadu, India
[2]Department of Disease Control, Faculty of Infectious and Tropical Diseases, London School of Hygiene & Tropical Medicine, London, UK
[3]Global Health Division, International Development Group, RTI International, Research Triangle Park, NC, USA
[4]Department of Global Health, University of Washington, Seattle, Washington, USA

**Acknowledgements** We thank field managers Rajeshkumar Rajendiran (Timiri) and Chinnaduraipandi (Jawadhu Hills), Jasmine Farzana, Gokila Palaniswamy, Janarthanan Maniyarasu, Naveen Kumar Sekar and Dhanalakshmi Manoharan (at CMC, Vellore) and David S Kennedy (at LSHTM) for their meticulous work in developing and implementing the coverage data collection forms. We are most grateful to our field supervisors and field workers in Timiri and Jawadhu Hills for exemplary work in completing the surveys despite the pandemic limitations. We acknowledge the contributions and efforts of the entire DeWorm3 Trials Team, particularly Dr Arianna Means, Dr Kristjana Ásbjörnsdóttir, and Emily Pearman at the University of Washington and Dr Tim Littlewood and Leanne Doran at the Natural History Museum. We are thankful for support extended to this study from the Ministry of Health and Family Welfare, Delhi and the Directorate of Public Health, Chennai, as well as help in implementing study activities from officials at the district and block levels, village leaders and members of our community advisory boards in Timiri and Jawadhu Hills. Most importantly, we thank all the participants for their time.

**Contributors** RMR and WEO conceptualised the study, conducted the data analysis, and wrote the first draft with support from SSRA and KEH. JW, the Deworm3 study PI, obtained funding for the study and helped write the paper. KA, SPK and GJI conceptualised the study, oversaw the analysis and provided feedback on the draft. SG and HL provided access to the data and supported critical interpretation of the findings. All authors (RMR, WEO, GJI, KA, SG, HL, SPK, JLW, KEH and SSRA) provided feedback on the manuscript and approved the final version. SSRA will be responsible for the overall content as the guarantor.

**Funding** The DeWorm3 study is funded through a grant to the University of Washington, Seattle, Washington, United States of America, from the Bill and Melinda Gates Foundation (OPP1129535, PI JLW). Under the grant conditions of the Foundation, a Creative Commons Attribution 4.0 Generic License has already been assigned to the Author Accepted Manuscript version that might arise from this submission. The funders were not involved in the decision to publish the manuscript and had no role in data collection, analysis or publication of study results.

**Competing interests** None declared.

**Patient and public involvement** Patients and/or the public were not involved in the design, or conduct, or reporting, or dissemination plans of this research.

**Patient consent for publication** Not applicable.

**Ethics approval** The DeWorm3 study was approved by the Institutional Review Board of Christian Medical College, Vellore, India (10392, dated 30 November 2016), Human Subjects Division at the University of Washington (STUDY00000180) and The London School of Hygiene & Tropical Medicine (12013) and conformed to the principles embodied in the Declaration of Helsinki. Participants gave informed consent to participate in the study before taking part.

**Provenance and peer review** Not commissioned; externally peer reviewed.

**Data availability statement** Data are available upon reasonable request.

**ORCID iDs**
Rohan Michael Ramesh http://orcid.org/0000-0002-1089-6037
William E Oswald http://orcid.org/0000-0002-5287-2834
Judd Walson http://orcid.org/0000-0003-4836-720X
Katherine E Halliday http://orcid.org/0000-0001-5307-2823
Sitara S R Ajjampur http://orcid.org/0000-0002-1089-6037

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
