## [Reviewer comments · BMJ Open]

ARTICLE DETAILS

TITLE (PROVISIONAL)	Representativeness of a mobile phone-based coverage evaluation survey following mass drug administration for soil-transmitted helminths: a comparison of participation between two cross-sectional surveys
AUTHORS	Ramesh, Rohan; Oswald, William; Israel, Gideon; Aruldas, Kumudha; Galagan, Sean; Legge, Hugo; PUTHUPALAYAM KALIAPPAN, SARAVANAKUMAR; Walson, Judd; Halliday, Katherine; Ajjampur, Sitara

VERSION 1 – REVIEW

REVIEWER	Scallan Walter, Elaine University of Colorado Denver - Anschutz Medical Campus
REVIEW RETURNED	31-Jan-2023

GENERAL COMMENTS	The paper compared the representativeness of a mobile phone-based coverage evaluation survey following mass drug administration for soil-transmitted helminths with coverage using the more usual approach of a in-person survey. Given the increase in cell-phone coverage in many countries and an increased interest in the use of telephone survey by public health practitioner, this is an important question. The paper is very well written, and the methods are sound. This will be of great interest to readers exploring the option of conducting telephone surveys in these settings. I have no additional comments or changes to suggest.
--

REVIEWER	Palmeirim, Marta University of Basel
REVIEW RETURNED	06-Feb-2023

GENERAL COMMENTS	I would highly recommend that someone with a strong background in statistics reviews the methodology and results section of the manuscript prior to the next round / submission.
--

REVIEWER	Verdonck, Kristien Institute of Tropical Medicine
REVIEW RETURNED	25-May-2023

GENERAL COMMENTS	The authors study to what extent the participants in a phone-based survey (due to COVID restrictions) and an in-person survey (6 months earlier) represent the target population in a specific setting. I consider the topic relevant and the information valuable. I particularly appreciate the quantitative documentation of non-representativeness due to non-participation with two survey modes (in-person and phone-based). This gives unique insight in origins
---

	and processes of bias which may vary across survey modes. On the other hand, I have several concerns about the manuscript which I list below. In addition, I have formulated a set of minor questions and suggestions in the annotated pdf file. Major comments  - The study is labelled as a cross-sectional study. However, the design turns out to be more complex than that. There appear to be two time points and several comparisons. Only after reading the results section, it became clear to me what the investigators had done exactly. I recommend to introduce the different comparisons (groups, denominators) more clearly in the methods. - The abstract does not clearly explain the study purpose and methods. The abstract should also provide a fair summary of the main findings; now only some of the findings are (over)emphasised. - There is a lot of generalisation in the conclusions (both in the abstract and the main paper). I recommend to keep the conclusions closer to the study. - The terminology of survey methods could be used more precisely. Is this study about non-response, non-participation, biased estimates of MDA coverage...? More reference to survey design methods could help to improve or streamline the wording (see specific comments in the annotated pdf file). - How were the trial treatment arms and the sampling frame managed in the statistical analysis? This requires more explanation. Have I understood correctly that only part of the statistical analysis takes the (two-staged?) sampling into account? - The model selection approach in the modified Poisson regression requires more explanation/justification. - It is not clear to me why the inclusion of mobile-phone related variables is called a sensitivity analysis? At first sight, this does not seem to assess the impact of a decision in the primary analysis plan. - The findings of table 3 are not sufficiently highlighted in the text. Would it be fair to say that this study documents representativeness issues with both survey modes - but that the size of the problem (and maybe sometimes also the direction) differs? - Discussion. I think that the recommendation to oversample in phone-based surveys is not in line with the findings of the study. Oversampling would not solve the problem of representativeness and could give a wrong impression of precision.
--	---

VERSION 1 – AUTHOR RESPONSE

Reviewer: 1

Dr. Elaine Scallan Walter, University of Colorado Denver - Anschutz Medical Campus

Comments to the Author:

The paper compared the representativeness of a mobile phone-based coverage evaluation survey following mass drug administration for soil-transmitted helminths with coverage using the more usual approach of an in-person survey. Given the increase in cell-phone coverage in many countries and an increased interest in the use of telephone survey by public health practitioner, this is an important question. The paper is very well written, and the methods are sound. This will be of great interest to

readers exploring the option of conducting telephone surveys in these settings. I have no additional comments or changes to suggest.

Thank you for reviewing our manuscript and the kind words of encouragement.

Reviewer: 2

Dr. Marta Palmeirim, University of Basel

Comments to the Author:

Dear authors,

This is a very interesting and important study - great job. Please do not forget to have a look at the changes I have suggested directly on the pdf.

All the best to all of you

We appreciate the thorough review of our manuscript and your positive feedback.

We have addressed your specific comments below and revised the manuscript where relevant. We have also incorporated your suggestions from the annotated pdf manuscript file into the latest version of our manuscript.

General comments

In this study the authors aimed at comparing the representativeness and implementation of a phone-based evaluation survey compared to an in-person survey for assessing the coverage of soil-transmitted helminth mass drug administration in India. This study has gathered important evidence that can serve as a basis for those conducting coverage surveys of any neglected tropical disease to make decisions on which methodology to choose based on the setting. Overall, I think this study should be published after some minor revisions. To enhance clarity and improve even further the manuscript, I made specific comments and suggestions which you can find below and in the edited version of the pdf (please do not forget to look at the pdf).

Specific comments:

1. Line 47 page 6: 109 what? Mobile phones? If yes I suggest "with 109 mobile phones per 100 people"

Thank you for your suggestion. We have now edited this sentence in the manuscript to report telephone connections per 100 people.

2. Line 49 page 6: 90% what?

We have now edited this, as already mentioned in the previous sentence.

3. Line 13 page 7: I removed this last part of the sentence as you say the same in the following sentence.

Thank you.

4. Line 47 page 7: How about school-aged children that do not attend school? Or do you perhaps mean pre-school aged and school-aged here?

We have now edited the sentence in the manuscript to specify both 'pre- and school-aged children'.

5. Line 31 page 8: What is the meaning of NDD and why is it mentioned after "school-aged children"?

National Deworming Day (NDD) is the national program for targeted deworming of pre- and school-aged children in India. We have now edited the introduction and methods, describing the trial and other background information to clarify this.

6. Line 17 page 9: Unclear what is "in Tiruvannamalai district" - are both sub-sites there?

Please rephrase for clarity

We have rephrased this sentence to clarify that the Jawadhu hills sub-site is located within the 'Tiruvannamalai district'.

7. Line 26 page 11: What do you mean by "Time of residence"?

Thank you. We have changed it to 'period of time in residence'.

8. Lines 6-7, page 13: "..., of the 2000 households sampled, ..." or "of the 2000 households selected"?

Thank you for your suggestion. We prefer to use the term 'sampled' instead of 'selected' because these 2000 houses were sampled from the entire population enumerated in the study's annual census.

9. Lines 15-25, page 13: I moved one sentence so that there is not such a back and forth between info about the phone-based, in person-based and then again phone-based surveys

Thank you. Revised as suggested.

10. Pages 23-24, second paragraph of the discussion: I feel this paragraph lacks some connecting sentences. It jumps very quickly from topic to topic without creating the link between the findings of the studies mentioned and the findings of the current study

Thank you. We have now added sentences that better connect the contents of the paragraphs.

11. Line 3, page 24: The references number 29 and 30 should be merged into one set of parenthesis no?

We have now merged them in the manuscript.

12. Lines 10-15, page 24: "Households with younger, more educated, and male heads of household were overrepresented in the phone-based survey." This sentence is too similar to that in lines 34-39 of the previous paragraph. Would be good to rephrase

As suggested we have rephrased the sentence to "The phone-based survey had a higher proportion of respondents from households with younger, more educated, and male heads of household compared to the censused population".

13. Lines 33-42, page 24: Could it also be because people do not have electricity to charge the battery of their phones or this is not a reality in this region?

Although there are occasional 'power-cuts' especially during the summer and rains, in the state of Tamil Nadu electricity is available most of the time and not so much of an issue.

14. Line 51, page 24: I would start a new paragraph after "In the in-person survey, most participants responded during the day..." and would probably add a few words before introducing this new topic of "best time to call"

Thank you for your suggestion, we have now made this edit in the manuscript.

15. Lines 40-45, page 25: But in an in-person survey this could be changed as well, making it possible for whoever is present to respond in the name of others, no? So I don't see this as a real advantage of the phone-based survey if you simply change this.

As mentioned in the methods section, the in-person survey was designed in a way that necessitated interviews with every household member. If unavailable, each participant was visited 3 times before the interview was terminated. Proxy responses were permitted only for children under 5 years of age.

However, in the phone-based survey, this could not be implemented. We have highlighted this not as an advantage but merely to explain why the phone-based interviews were faster.

16. Line 54-56: This last sentence feels a bit lost here. Could it be moved up to were you mention the possibility of a bias introduced due to proxy responses and merge both? As suggested by another reviewer we have now omitted this sentence.

Reviewer: 3

[PLEASE SEE ATTACHED FILE (bmjopen-2022-070077_Proof_hi_tv.pdf) FOR ADDITIONAL COMMENTS FROM REVIEWER 3]

Dr. Kristien Verdonck, Institute of Tropical Medicine

Comments to the Author:

The authors study to what extent the participants in a phone-based survey (due to COVID restrictions) and an in-person survey (6 months earlier) represent the target population in a specific setting. I consider the topic relevant and the information valuable. I particularly appreciate the quantitative documentation of non-representativeness due to non-participation with two survey modes (in-person and phone-based). This gives unique insight in origins and processes of bias which may vary across survey modes. On the other hand, I have several concerns about the manuscript which I list below. In addition, I have formulated a set of minor questions and suggestions in the annotated pdf file.

Thank you for the extensive and detailed comments and suggestions that have helped to improve our manuscript. We have addressed the specific comments from the annotated pdf manuscript file below and made changes in the manuscript as well.

General comments (Major comments)

1. The study is labelled as a cross-sectional study. However, the design turns out to be more complex than that. There appear to be two time points and several comparisons. Only after reading the results section, it became clear to me what the investigators had done exactly. I recommend to introduce the different comparisons (groups, denominators) more clearly in the methods.

Please see our response below in comment no. 6 in the specific comments section.

2. The abstract does not clearly explain the study purpose and methods. The abstract should also provide a fair summary of the main findings; now only some of the findings are (over)emphasised. Please see our response below in comment no. 3, 5, 7, 8, 10, and 12 in the specific comments section.

3. There is a lot of generalisation in the conclusions (both in the abstract and the main paper). I recommend to keep the conclusions closer to the study.

Please see our response below in comments no. 14, 16, 27, 69, and 72 in the specific comments section.

4. The terminology of survey methods could be used more precisely. Is this study about non-response, non-participation, biased estimates of MDA coverage...? More reference to survey design methods could help to improve or streamline the wording (see specific comments in the annotated pdf file).

Please see our response below in comment no. 40 and 41 in the specific comments section.

5. How were the trial treatment arms and the sampling frame managed in the statistical analysis? This requires more explanation. Have I understood correctly that only part of the statistical analysis takes the (two-staged?) sampling into account?

Please see our response below in comment no. 42 in the specific comments section.

6. The model selection approach in the modified Poisson regression requires more explanation/justification.
7. Please see our response below in comment no. 42 and 44 in the specific comments section.
8. It is not clear to me why the inclusion of mobile-phone-related variables is called a sensitivity analysis. At first sight, this does not seem to assess the impact of a decision in the primary analysis plan.
Please see our response below in comment no. 45 in the specific comments section.
9. The findings of table 3 are not sufficiently highlighted in the text. Would it be fair to say that this study documents representativeness issues with both survey modes - but that the size of the problem (and maybe sometimes also the direction) differs?
Please see our response below in comment no. 11 and 12 in the specific comments section.
10. Discussion. I think that the recommendation to oversample in phone-based surveys is not in line with the findings of the study. Oversampling would not solve the problem of representativeness and could give a wrong impression of precision.
Please see our response below in comment no. 61 in the specific comments section.

Specific comments (from annotated pdf manuscript file):

1. Line 3 Page 3 : Does this refer to the coverage of the survey (i.e. "frame" or "coverage" error) or to the coverage of the mass drug administration? I think it is important to make that clear in the beginning of the manuscript.

Thank you for your query. As defined by WHO "Coverage evaluation surveys are straightforward, population-based surveys designed to provide precise estimates of preventive chemotherapy coverage against targeted NTDs and provide a valuable tool for evaluating programmatic performance." We have now explained this more clearly in the abstract and introduction.
<https://apps.who.int/iris/bitstream/handle/10665/329376/9789241516464-eng.pdf>

2. Line 7 Page 4 : There seems to be a mismatch between the titles and the content of the subsections in the abstract.

We have now modified the contents of the abstract to align with the title of the study.

3. Line 16 Page 4 : After reading the introduction, I expected a before-after study. After reading the abstract, I do not understand yet how the COVID pandemic influenced this study.

Thank you for your comment. We have now modified the objectives in the abstract and the last paragraph in the introduction to explain the study design more clearly and how the pandemic lockdown necessitated the use of a phone-based coverage evaluation survey when it was previously conducted in-person.

4. Line 21 Page 4 : This study was done within the setting of a trial. How could this influence the findings? Is this discussed as a limitation?

Thank you for your comment. We do not anticipate any influence of the trial setting on the findings of our study. In contrast, the current study was only possible because of the trial setting, which permitted us access to a complete and detailed census that allowed us to assess the representativeness of the participants in the phone-based versus in-person surveys. The trial setting is also a strength as standardised data collection ensured data quality for the proposed objectives. It is possible that overall participation in the coverage evaluation surveys conducted as part of the DeWorm3 trial could be higher (or lower) than what might be observed in a programmatic setting, as a result of presence

and recognition in the communities, but we do not see the trial setting having an impact on the comparison we make within the trial setting between survey methodologies. We have now described this in the discussion section.

5. Line 23 Page 4 : What are the intervention arms of this trial? What is the purpose of the trial?
Thank you. The intervention arm of the trial was twice-yearly cMDA for three years. We have now described this in the manuscript as well.

6. Line 25 Page 4 : Is this a cross-sectional design? Do you compare different cross-sectional surveys which were organised at different points in time? Does this study include a before-after evaluation? I recommend to reflect on the description of the overall study purpose and design. Only in the results section, it becomes clearer what exactly this study entails/compares.

Thank you for your queries. We compared two cross-sectional surveys organised at different points (6 months apart) using two different data collection methods and is not a before and after evaluation. The type of study design is now changed to 'A comparison of participation between two cross-sectional surveys'. We have now explained this in detail in the methods section.

7. Line 29 Page 4 : So all households were approached twice? Which mode came first?

As suggested earlier, in the objectives of the abstract, we have now incorporated a sentence to explain that the phone-based coverage evaluation survey was implemented (due to an abrupt lockdown during the COVID-19 pandemic), as an alternative to the routine in-person surveys conducted previously.

8. Line 41 Page 4 : Why is there no methods section in the abstract? And what was the purpose of this study? The objectives section reads as if it was about estimating treatment coverage, but further down it is all about nonresponse (or nonparticipation?) in different survey modes and associated factors.

Thank you for the query. We drafted the structured abstract as per the guidelines of the journal to include objectives, design, setting, participants, intervention (if applicable), outcome measures, results, conclusion and trial registration. We will attempt to incorporate further details of the 'methods' under the design section if we are able to comply to the word limit of 300 words. We have revised the abstract to state that we aimed to compare representativeness and implementation (including non-response) of these two survey modes through a comparison of two cross-sectional surveys.

9. Line 44 Page 4 : Please reformulate. The sentence suggests that the 346 households without phone number were part of the 57% who participated. I suppose this is not correct?

Thank you for pointing this out. We have now corrected this sentence in the manuscript.

10. Line 44 Page 4 : What is the focus of this study? Phone ownership, willingness to participate in a phone-based survey, or both? What happened with the 26% (correct?) who did provide a phone number but did not participate during the phone survey: were they not reached or did they refuse to participate?

Thank you for your comment. The study focuses on comparing two commonly used survey modes, (in-person and phone-based). As suggested, both phone ownership and willingness to participate in a phone-based survey are important to ascertain participation and representativeness. Due to restrictions in the word limit in the abstract we have now mentioned this in more detail in the introduction section. As illustrated in Figure 2, 25.3% of households (505/2000) who had provided phone numbers could not be contacted (numbers were unreachable, call not picked up and out of town, and wrong number). Verification by visiting the houses was not feasible as there was a travel restriction imposed nationwide due to the lockdown. We hope the content in the text explains this in more detail.

11. Line 51 Page 4: On which analysis is this statement based? It overlaps partially but not completely with the findings of the regression analysis. I suggest to include only the final results in the abstract. Or to explain why you present two sets of factors associated with nonresponse.

Thank you for your query. The comparison of household characteristics between censused and participating households of the in-person and phone-based coverage surveys have been presented in Table 1. We have highlighted only the significant differences in characteristics between censused and participating households from this as 'smaller households, households with lower socioeconomic status, and those with older, female or less educated household heads were under-represented in the phone-based survey compared with the censused population'. The next sentence highlights the key findings from Table 2 which is the modified Poisson regression for factors associated with non-response in the phone-based survey.

While we would like to highlight findings from the other analyses, we are unable to due to the journal's 3000 word limit and chose to highlight those indicating the level of representativeness and non-response of the phone-based survey.

12. Line 52–53 Page 4 : Figure 2 and table 3 appear to be absent from this results summary. Thank you for your query. Due to constraints in the word limit in the abstract we chose not to include most details from Figure 2 which illustrated participation in both surveys and Table 3, which presented the results of our model of non-response during the in-person survey.

13. Line 3 Page 4 : Consider using a different word. A regression analysis helps to estimate something under a set of assumptions it does not "determine"...

Thank you. We have now used the word 'revealed' instead.

14. Line 8 Page 5 : These conclusions imply a lot of generalisation: from this trial setting in India to all surveys, from STH to all NTD, from nonresponse to biased coverage estimates and to impacting those who most need treatment. I recommend to keep the conclusions closer to the findings of this study in a particular setting.

Thank you. We have revised the abstract's wording and conclusion.

15. Line 10 Page 5: To what extent is this statement based on the results?

Thank you for your query. This statement is based on the fact that STH is prevalent in tropical countries specific to less privileged households/individuals and cMDA intends to target such groups. It is imperative to ascertain true treatment coverage estimates, especially in these less privileged groups. As our study suggests, if the coverage evaluation survey is phone-based, these groups could be missed or underrepresented thus skewing estimates. We hope the content in the text explains this in more detail.

16. Line 13 Page 5: What I find valuable in this study is the empirical study of non-representativeness due to non-participation with two survey methods (in-person and phone-based) in one setting. This gives unique insight in origins and processes of bias which may differ depending on the survey mode. I suggest to highlight this better rather than (over) generalising the findings to all phone-based surveys, all NTDs, and the whole world.

Thank you. We have revised the abstract following your suggestions and highlighting this strength of our study.

17. Line 15 Page 5: I do not see the link between this statement and the results section of this abstract.

Thank you for your comment. We have removed this statement from the abstract.

18. Line 29 Page 5: This should be included in the abstract. It is a key part of the methods.

Thank you for this suggestion. We have now included this in the abstract.

19. Line 31 Page 5: This first part of the second bullet point overlaps with the first bullet point. Thank you, we have now modified the first and the second bullet points to ensure no overlap in their contents.

20. Line 35 Page 5: This would be relevant if you would compare the participants' replies to the questions, but based on the abstract, it seems to be all about participation, not the content of the replies? I wonder why the information about MDA coverage is not included in this paper? That would be necessary to empirically assess the coverage bias.

Thank you for your query. Yes, we have not used the content and individual responses of the questionnaires. A broader analysis of treatment coverage across the DeWorm3 trial sites and all six MDA rounds is forthcoming, which limited the information on MDA coverage that we could describe here. However, we choose to highlight the strength that the survey questionnaire was identical between rounds. In contrast to several previous studies that attempted to compare implementation using different questionnaires at different time points, the structure, length, and questions were the same in both our surveys, which ensured very little variation in the way they were administered after extensive training. Using the meta-data associated with each questionnaire response, we were then able to compare duration, individuals completed per visit or call, and total time to complete the household's survey.

21. Line 48 Page 5: And what about the fact that the study occurred within a trial?

Thank you for your comment. As mentioned earlier in comment no. 4, the objectives of this study are specific to the activities implemented within the trial so we are quite certain that the trial does not influence the findings of this study. In fact, since this study is implemented within a trial, it offers standardised data collection and ensures data quality. The trial also provided a censused population from which the participants were randomly selected.

22. Line 53 Page 5: How do you know the direction of the impact? Could it also have been in the other direction (more people participating because they had more time during lockdown)?

Thank you for your suggestion. We have now removed the word 'negatively' in the manuscript.

23. Line 13 Page 6: assessing?

Thank you for pointing this out. We have changed this in the manuscript

24. Line 22 Page 6: A more precise term could be "survey mode". The terms "survey method" and "survey strategy" seem to be too broad.

Thank you for your suggestion. We have now changed this throughout the manuscript.

25. Line 24–25 Page 6: Is this an advantage?

Thank you. We have now removed this statement.

26. Line 31 Page 6: Why would this be specific to phone-based surveys?

Thank you. We have now removed this from the manuscript and highlighted the exclusion of participants without access to phones as a disadvantage.

27. Line 38 Page 6: Different terms are used. What is your main concern? Biased (i.e. invalid) survey findings due to non-participation? Hence wrong ideas about MDA coverage (overestimation or underestimation of the MDA coverage)? I suggest to explain that clearly and stick to a set of key terms. I think that the term "generalisability" is too broad in this context.

Thank you. We have now removed the term "generalisability".

28. Line 42 Page 6: Is this a mean for the country?

Thank you. We have now clarified it as the mean teledensity for the country.

29. Line 49 Page 6: Is it 89% or 90%? Please reformulate clearly and avoid (near) overlaps.

Thank you for pointing this out. We have now clarified the terminology and estimates for teledensity in both the sentences.

30. Line 12 Page 7: Filariasis is a disease, not a programme. I suggest to reformulate this.

Thank you. We have removed 'lymphatic filariasis' from this sentence as the control program for it is better described in the next sentence.

31. Line 31 Page 7: Does this imply that India does not follow the recommendations for pre-school-aged children and women of reproductive age?

Thank you for your query. In the NDD program, India recommends biannual deworming of children between 1 to 19 years of age, and women of reproductive age are not included. For clarity, we have now removed this text from the introduction and moved the description of NDD to the methods.

32. Line 32 Page 7 : Does this refer to the WHO tool? Possible to drop these two words?

Thank you. We have now removed 'as standard' from this sentence.

33. Line 8–9 Page 8 : When were these in-person surveys done?

Thank you for the suggestion. We have now modified this paragraph to highlight that the previous four coverage evaluation survey rounds were conducted in-person and the fifth round necessitated a phone-based survey due to the restrictions of the lockdown imposed in response to the pandemic.

34. Line 45 Page 8: The survey was also done in the control clusters even if they did not receive cMDA? Does this imply that cMDA was organised in the same week in all clusters?

Thank you for the query. Yes the coverage evaluation survey was also done in the control clusters even if they did not receive cMDA, to assess the coverage of the school-based national deworming day program. As you have mentioned, cMDA was organised in the same week simultaneously in all clusters. cMDA is implemented the day after the national deworming day, with mop up rounds revisiting previously unavailable houses that last a week. Coverage evaluation surveys were conducted in the intervention and control clusters within a week of the cMDA. We have now added these details to the methods.

35. Line 32 Page 8: So this applies both to the phone and the in-person surveys?

Thank you for your query. Data from both the phone and the in-person surveys were collected using the same electronic data collection forms. We have now mentioned this in the methods section in the manuscript.

36. Line 51 Page 9: Only now it becomes clear that you compared one in-person round with one phone-based round which are 6 months apart and use two different samples based on two different census rounds. I think you could have explained this higher up.

Thank you for your suggestion. We have now explained this ahead in the last paragraph of the introduction section.

37. Line 23 Page 10: Did the COVID-19 pandemic influence incoming or outgoing mobility in the study area?

Thank you for your query. Strict travel restrictions imposed during the complete national lockdown due to the COVID-19 pandemic ensured very minimal incoming or outgoing mobility in the study area.

38. Line 10 Page 11: Which households are included here? All those that were not randomly selected? Why does it say "among those" instead of "all"?

Thank you for your query. We have revised the wording of this section for clarity.

39. Line 12 Page 11: variables/factors associated with...?

Revised as suggested.

40. Line 15–19 Page 11: I do not understand this. Please clarify or provide more information.

Thank you for your query. The intended sample size for each of the coverage evaluation surveys was 2000, so we first selected 2000 households from the census but additional samples were also randomly selected to compensate for non-response. Typically, in each coverage evaluation survey, after approaching/contacting all the households in the first list, these additional lists of replacement households were released systematically. Replacement lists were released until the final sample size of 2000 was achieved in each survey. However, for this analysis we considered and compared only the details from the primary list of 2000 households in both surveys. We have provided more detail in this section for clarity.

41. Line 24 Page 11: Are the variables that follow the characteristics that are compared? But if that is the case, why does the text say "based on"?

Thank you for your comment. We have now replaced 'based on' with 'considering'.

42. Line 38 Page 11: How was the two-stage (?) survey design (clusters - randomly selected households - all people within the selected households) managed in the analysis?

The household-level analysis using modified Poisson regression accounted for clustering of households within the trial's clusters using robust errors. This is described in the second paragraph under 'data analysis'. The statistical tests to compare surveyed households with non-surveyed households using the census and the visit-level analysis to compare implementation did not require the survey design to be incorporated in the analysis.

43. Line 10 Page 11: How does this differ from "time to complete the survey" and "time of calls"?

Thank you for the query. We have now modified the manuscript to explain the difference between the timing of the calls (time of the day the calls were made), duration per visit or call, and duration to complete the household's survey.

44. Line 17–24 Page 12: Why this approach? I recommend to justify this choice.

There are multiple approaches to model-building. For this study for which we had little prior knowledge of potential drivers, we felt that an approach examining all possible combinations of a host of candidate predictors towards maximizing model fit based on the Bayesian Information Criterion (BIC) was a suitable and pragmatic approach, considering the size of our data and number of candidate predictors (Heinze et al 2017).

45. Line 29 Page 12: Why is this a sensitivity analysis? At first sight, it does not seem to check the impact of a decision in the primary analysis plan?

While not a pre-specified analysis, we implemented this sensitivity or 'stability' check exactly to examine the impact of including two alternate, and likely highly influential, predictors in our model-selection procedure, specifically reported phone ownership and phone number availability. Phone ownership is potentially driven by socioeconomic status and education, and indeed when we included phone ownership in the selection procedure, education was no longer selected, implying that its association was weakened to the point of non-significance. Similarly, the association of SES was attenuated but not removed completely with the inclusion of mobile phone ownership, perhaps indicating that there is an open pathway between SES and non-response (as the reviewer indicated - some unmeasured factor). Alternatively, there could be misclassification of the ownership variable

between its measurement during the census and the time of the subsequent survey, whether by accident, actual change, or intention by the respondent. The impact of including the availability of a phone number is potentially an even more proximal factor to the outcome compared to reported phone ownership, as indicated by the stronger association, but it did not further change the other variables selected. Exploring these alternate measures provided useful insight into the structure of the relationships between the candidate predictors and our outcome, which may prove useful for subsequent research in this area.

46. Line 29 Page 12: Was a data analysis plan (publicly) available before the start of the analysis? Is the whole study a secondary analysis of the trial data that were collected anyway?

Thank you for your query. Data from the trial cannot be shared publicly because the study remains blinded to outcome data. However, following unblinding of the study, data will be available upon reasonable request. This analysis was not planned a priori but was possible due to the change in survey mode necessitated by the complete nationwide lockdown in response to the COVID-19 pandemic.

47. Line 15 Page 13: This figure contains important findings of this study. Would it be possible to mention more of that in the abstract?

Thank you. We agree with you, but due to restrictions in word count, we are unable to currently incorporate this in the abstract.

48. Line 4 Page 13: Sounds strange; I suggest to reformulate.

Thank you for the suggestion. We have now rephrased the sentence.

49. Line 32 Page 15: How could you explain that 43 households who reported not to own a mobile phone were reached by (mobile) phone?

Thank you for this pertinent query. Details for these 43 households were extracted from the prior census. They reported not owning a cell phone during the census, however, they provided alternate phone numbers of neighbours /relatives through which we contacted them for the phone-based coverage evaluation survey.

50. Line 55 Page 16: The type and denominator of the comparison groups varies across this paper. Possible to make that clearer in the methods section?

Thank you for your query. To assess factors associated with non-response in both surveys we compared characteristics of the non-response households and household heads to the primary sampled coverage evaluation households in both surveys. Though we intended to include details of all 2000 households in both surveys, the denominators varied since we excluded the primary sampled coverage evaluation households with missing covariate data (illustrated in Figure 2 and highlighted in the footnotes). We have now explained this in the methods section of the manuscript as well.

51. Line 3 Page 19: The findings of table are also important and do not come out clearly in the abstract, introduction, methods. There are issues with representativeness with the two survey modes - but the size of the bias (and maybe sometimes also the direction) varies.

As described above, we were unable to present results from table 3 in the abstract because of word limit prescribed by the journal. We described the results of the in-person survey model in the results.

52. Line 43 Page 20: I do not think that this section adds much. It is not surprising that some of these variables are intercorrelated? It could be good to acknowledge in the limitation section that there may be intercorrelation, and that the associated factors/variables identified in his study may well be proxies for other unmeasured factors. I also assume that you have not checked for effect modification/interaction? Taken together, this study cannot identify "determinants", only "associated/correlated factors".

The reviewer is correct that we have not taken a causal approach to our analysis. We have employed various approaches to examine representativeness and non-response in our study and further used sensitivity analyses to test our modelled conclusions when other candidate predictors were considered. Our study exploited the opportunity afforded by the change in trial protocol necessitated by the national lockdown to explore factors associated with non-response to a phone and in-person survey and characterise the representativeness of the surveyed population. As such it was limited in the data available and there may be unmeasured factors. We have acknowledged this now in the discussion, but we feel our conclusions are suitably measured and useful for other research in this area.

53. Line 20 Page 22: Is this the number (proportion) of visits where the household was not located, where all HH members had deceased, where nobody was present, where at least one household member was present? Would it be possible to make that clearer in the table or legend?

The text in the table has now been revised to clarify that this is the "Household status during visit or call".

54. Line 38 Page 22: Is this the median number of individuals with whom the interviewer talked directly, per attempt (visit or call)?

Yes, this refers to the median number of resident interviews completed per visit or call. The text in the table has now been revised to clarify this.

55. Line 40 Page 22: What does this mean?

The text in the table has now been revised to clarify this.

56. Line 20 Page 23: This term seems too broad. You only looked at a few aspects of the implementation. Would it be possible to find more specific terms?

Thank you. Revised as suggested.

57. Line 39 Page 23: Only for sub-site, or are these the findings of the multi-variable analysis that also includes other factors?

Yes, these are findings from the multivariable analysis of selected candidate predictors.

58. Line 53 Page 23: "those strata" or "that stratum"

Thank you for pointing this out. We have changed this in the manuscript.

59. Line 26 Page 24: This sounds awkward. What do you mean exactly?

Thank you for your comments. A report by Harvard Kennedy school on "Understanding barriers to and impacts of women's mobile phone adoption in India"

(<https://research.hks.harvard.edu/publications/getFile.aspx?Id=2765>) stated that the main reason for poor adoption of mobile phone among women in India is technical illiteracy (unable to effectively operate or make use of the various functions, settings, and applications available on a mobile phone due to a lack of familiarity or understanding). We have used now used the same term 'technical illiteracy' in the manuscript as well.

60. Line 33 Page 24: Insert 'had'

Thank you for your suggestion.

61. Line 59 Page 24: This sounds inappropriate after you have just shown that there is a problem with representativeness. Starting with a larger target sample does not solve bias! It may even give the wrong impression that the estimates will be fine and precise while in reality they are as biased as with a smaller sample size. This recommendation is inconsistent with the findings of this study!

Thank you for the comment. We have now modified the sentence and removed our recommendation for oversampling. As suggested later, we have also moved our recommendation on quota sampling here from the conclusion section.

62. Line 17–19 Page 25: That depends on the setting. A more appropriate recommendation could be to check when the target population is most likely to respond to calls. But then, I suppose that it is also recommended to check when the target population is most likely to open the door for an in-person survey....

Thank you for the comment. We have now modified this statement to consider, the local culture and work patterns of the target population while planning survey timing to ensure high response rates. We have also highlighted in the following statement that phone-based surveys provide the advantage of flexibility in terms of survey timing, enabling data collection beyond the typical working hours and daylight constraints associated with in-person surveys, particularly in the context of safety considerations in many countries.

63. Line 22 Page 25: Insert arrange / set / organise

Thank you for your suggestion. We have now modified this statement accordingly.

64. Line 33–35 Page 25: That is the elephant in the room of course. Why would you write "may increase". Getting to the houses will of course take much more time than calling people on the phone. Is the duration of the interview even relevant in this comparison?

Revised as suggested to replace "may" with "would".

65. Line 47 Page 25: This is now about another type of bias in surveys, i.e. measurement bias (or information bias). There are many more things to say about measurement bias in function of the survey mode. I suggest to either discuss this more in-depth or to focus on representativeness only. Thank you for the suggestion. We have now removed this statement to focus only on representativeness.

66. Line 54 Page 25: remove period

Thank you for the suggestion.

67. Line 54–56 Page 25: Overlap with previous comment about measurement bias.

Thank you for pointing this out. We have now removed this statement.

68. Line 17–23 Page 26: Too many things together? Please explain better. Is the estimated MDA coverage an outcome of the trial? Biased estimates of MDA coverage would then invalidate (part of) the trial. Biased estimates of MDA coverage in a survey can be due to nonresponse, non-participation and measurement errors. The mode of the survey (in-person or phone) can influence the risk of biased estimates in different ways (via response, participation, truthfulness of the answers to the survey questions). How do biased estimates of MDA coverage influence risk of STH infection (in a trial setting)? And how does all this relate with being at high risk of carrying STH is not straightforward (and outside the scope of this study). I consider that speculating on this reduces the strength of the argumentation in this paper.

Our results show that lower SES households were systematically missed during the phone-based survey. Baseline results from the trial and others sources indicated that these households were more likely to have higher STH risk. Our conclusion that the use of a phone based survey is not recommended because of the exclusion of these households is based on the premise that an accurate measure of coverage in those at-risk is critical to successful elimination of STH as a public health problem.

69. Line 36–37 Page 26: What follows adds to the internal validity of this study. It also reduces the external validity because as you explain, this is a rather unique setting (and different from other places). I suggest to formulate the supposed "generalisability (or external validity)" of this study more carefully. Cell phone ownership and participation in surveys are phenomena that are likely to be context-specific anyway? On the other hand, I do consider that empirically identifying processes of bias and learning from that is relevant across settings.

Thank you. Though this is specific to Tamil Nadu, an Indian state with high teledensity, we can confidently generalize the findings of this analysis to rural areas in other states of India with lower teledensity. The disparity in representativeness would likely be far worse in settings where phone coverage is probably lower.

70. Line 40 Page 26: The embedding of this study in a trial deserves more discussion.
Thank you. As discussed above in point no. 4 and 21.

71. Line 42 Page 26: I suggest to add the time between the two survey rounds. How much is reasonable?
Thank you for the suggestion. We have now provided the exact time period between the two surveys.

72. Line 8–9 Page 27: I consider that this generalisation goes too far.
We have now revised this in the manuscript

73. Line 22 Page 27: I suggest to move this idea from the conclusion to the discussion. Could stratified sampling also be a solution in this case (given the census information)?
Thank you for the suggestion. We have now moved this sentence to the discussion section in the manuscript.

74. Figure 1: This figure is very helpful to understand the setup of this study. I suggest to review the text (from the abstract onwards) and try to make this setup clearer in the wording.
Thank you for the suggestion. We have now explained this in more detail in the manuscript.

VERSION 2 – REVIEW

REVIEWER	Palmeirim, Marta University of Basel
REVIEW RETURNED	13-Sep-2023

GENERAL COMMENTS	Please find very minor suggestions of changes and 2-3 comments in the pdf attached below. Otherwise, I think this manuscript is ready for publication!
--

REVIEWER	Verdonck, Kristien Institute of Tropical Medicine
REVIEW RETURNED	09-Aug-2023

GENERAL COMMENTS	I appreciate the careful revision of the paper and the replies to my specific comments. I suggest to check the tense of the verbs in the "strengths and limitations" section: there is a mix of past and present tense ("allowed" and "allows"). Apart from that, I have no further suggestions.
--

VERSION 2 – AUTHOR RESPONSE

Reviewer: 3

Dr. Kristien Verdonck, Institute of Tropical Medicine

Comments to the Author:

I suggest to check the tense of the verbs in the "strengths and limitations" section: there is a mix of past and present tense ("allowed" and "allows").

Response: [Addressed]

Reviewer: 2

Dr. Marta Palmeirim, University of Basel

Specific comments:

1. Line 19 page 7: so only 1-19 year old children in school receive it in the control groups? Or also those not attending school? please clarify here

Response: Thank you for your suggestion. We have now edited this sentence in the manuscript to explain that only the children attending schools and preschools for 1 to 19 years were treated in the control arm.

2. Line 12 page 10: sometimes you write 2,000 and sometimes 2000 - make sure to harmonize prior to the next submission

Response: We have now edited this in the manuscript.

3. Line 56 page 15: Slightly different formatting, please harmonize: Table 1 - or Table 2: ?

Response: We have now edited this in the manuscript.

4. Line 29 page 25: Although if "one participant could respond on behalf of those absent," we are not quite measuring the same anymore, as one participant might not recall as accurately what others did - could this be mentioned as a potential limitation?

Response: Thank you for the suggestion. Yes, we agree with you. In the phone-based survey, many responses were proxy responses on behalf of the other members of the household. However, as described in the manuscript, we have highlighted it as a limitation of the phone-based survey approach rather than a limitation of the present analysis.